# Cross-Sectional Survey on Long Term Sequelae of Pediatric COVID-19 among Italian Pediatricians

**DOI:** 10.3390/children8090769

**Published:** 2021-08-31

**Authors:** Giuseppe Fabio Parisi, Lucia Diaferio, Giulia Brindisi, Cristiana Indolfi, Giuseppina Rosaria Umano, Angela Klain, Giuseppe Marchese, Daniele Giovanni Ghiglioni, Anna Maria Zicari, Gian Luigi Marseglia, Michele Miraglia del Giudice

**Affiliations:** 1Department of Clinical and Experimental Medicine, University of Catania, Via Santa Sofia 78, 95123 Catania, Italy; gf.parisi@policlinico.unict.it; 2Department of Pediatrics, Giovanni XXIII Hospital, University of Bari, 70126 Bari, Italy; luciadiaferio83@gmail.com; 3Department of Pediatrics, Allergology and Immunology Division, Sapienza University, Viale Regina Elena 324, 00161 Rome, Italy; giulia.brindisi@uniroma1.it (G.B.); annamaria.zicari@uniroma1.it (A.M.Z.); 4Department of Woman, Child and Specialized Surgery, University of Campania “Luigi Vanvitelli”, 80138 Naples, Italy; cristianaind@hotmail.com (C.I.); giusi.umano@gmail.com (G.R.U.); klainangela95@gmail.com (A.K.); 5Primary Care Pediatrician, ATS Montagna, 25051 Brescia, Italy; giumarche@hotmail.com; 6Fondazione IRCCS Ca’ Granda Ospedale Maggiore Policlinico di Milano, Via Francesco Sforza, 28, 20122 Milan, Italy; daniele.ghiglioni@policlinico.mi.it; 7Department of Pediatrics, Foundation IRCCS Policlinico San Matteo, University of Pavia, 27100 Pavia, Italy; gl.marseglia@smatteo.pv.it

**Keywords:** children, coronavirus, COVID-19, long COVID, sequelae, post COVID, survey, pediatricians

## Abstract

The persistence of symptoms after recovery from Coronavirus 2019 (COVID-19) is defined as long COVID, an entity that had occurred among adults but which is not yet well characterized in pediatric ages. The purpose of this work was to present some of the data from a survey addressed to Italian pediatricians concerning the impact of long-COVID among children who recovered from severe acute respiratory syndrome coronavirus-2 (SARS-CoV-2) infection. The questionnaire was designed and pre-tested in February 2021 by a working group of experts from the Italian Pediatric Society for Allergy and Immunology (SIAIP). The survey was emailed once in March 2021 to a sample of Italian pediatricians. A total 267 Italian pediatricians participated in our survey. According to most pediatricians (97.3%), the persistence of symptoms is found in less than 20% of children. Specifically, with regard to the symptoms that persist even after swab negativization, fatigue was the most mentioned one (75.6%). Long-COVID would seem to be a phenomenon of limited occurrence in pediatric ages, affecting less than 20% of children. Among all of the symptoms, the one that was most prevalent was fatigue, a pathological entity that is associated with many viral diseases.

## 1. Introduction

It is now known that many patients who recover from the acute phase of the Coronavirus disease 2019 (COVID-19) continue to have clinical manifestations or develop new ones. This finding has alerted the scientific community, and researchers immediately began investigating these alterations and the possible correlation with severe acute respiratory syndrome coronavirus 2 (SARS-CoV-2) infection. The term “long-COVID” was coined to indicate these manifestations [1].

To better define the discussion in this study, we referred to the guidelines produced by the National Institute for Health and Care Excellence (NICE), published on 18 December 2020 [2].

These guidelines use several clinical definitions:

- Acute COVID-19: signs and symptoms of COVID-19 up to four weeks after the onset of the disease.

- Ongoing symptomatic COVID-19: signs and symptoms of COVID-19 four to 12 weeks after the onset of the disease.

- Post-COVID-19 syndrome: signs and symptoms that continue or develop after an infection compatible with COVID-19, persist for more than 12 weeks from the onset of the disease, and are not explained by alternative diagnoses [2].

The NICE guidelines report that in addition to the previous clinical definitions, the term long-COVID is commonly used to describe the signs and symptoms that continue or develop after the acute stage of COVID-19, thus including both COVID-19 that continues to be symptomatic and the post-COVID-19 syndrome as defined above [2].

As for adult patients, much scientific evidence is already available. In this sense, the systematic review and meta-analysis by Lopez-Leon et al. highlights how 80% of patients who contracted COVID-19 developed at least one long-term symptom of which the five most common were fatigue (58%), headache (44%), attention disturbance (27%), hair loss (25%), and dyspnea (24%) [3].

As for children, the situation is slightly different. In fact, it is known that the course of COVID-19 in children is much less severe than in adults, and more serious complications, such as pneumonia, are less frequent [4,5,6]. However, recent evidence seems to show that long-COVID symptoms also affect younger patients [7,8,9,10].

In 2020, our group, which is part of the Italian Paediatric Society for Allergy and Immunology (SIAIP) Rhino-sinusitis and Conjunctivitis Committee, published the results of a survey that collected the responses from 99 Italian pediatricians concerning the impact of COVID-19 and the most frequently found clinical manifestations in affected Italian children [11]. Recently, it was decided to repeat the study of the previous year by involving a greater number of pediatricians in another survey to verify the way in which the scenario of COVID-19 clinical manifestations has changed one year after the previous pandemic peak. Furthermore, in this survey we decided to add questions that investigated the presence of long-term sequelae in pediatric patients who have had COVID-19.

The purpose of this article was therefore to present the results of this survey that evaluated the long-term impact of SARS-CoV2 infection in pediatric aged population to try to estimate the long-COVID phenomenon in this age group as well.

## 2. Materials and Methods

The questionnaire was designed and pre-tested in February 2021 by a working group of experts of Italian Paediatric Society for Allergy and Immunology (SIAIP). The survey was emailed once in March 2021 to a sample of about 500 Italian pediatricians. The questionnaire was structured into different sections of 35 categorized and multiple choice questions. The questions in the questionnaire focused on the clinical and management aspects related to the COVID-19 pandemic in pediatric age with its relative impact among Italian pediatricians. In this article, the data relating to the questions concerning the long-term sequelae, typical of the so-called long-COVID, are presented. Participants were allowed to complete only a single survey. The previously revised and confirmed paper version of the questionnaire was finally converted into a web-based survey with Google-Drive (Google DriveTM, © 2021 Google Inc. all rights reserved), a free internet platform applied for the creation of internet-based survey forms that allows real-time digital archiving of collected data, real-time presentation of survey results, and simple downloading of all data of registered anonymized participants in an Excel© format for statistical analysis. The language of the questionnaire was the national one. The reported time to complete the survey was approximately 15 min. Once the questionnaire results were obtained, they were statistically processed. The Ethics Committee of the University of Catania (Italy), was contacted and no special permission was deemed to be required because the study design satisfied the criteria of an activity audit.

The survey questions in this article are shown in Table 1.

## 3. Results

A total of267 Italian pediatricians participated in our survey and provided responses to our electronic questionnaire by 31 March 2021. The characteristics of the survey participants are detailed in Table 2. From the characteristics of the participants, it is clear that most of the participants were primary care pediatricians over the age of 60 who practice their profession in Northern Italy.

As for the answers to the selected questions, according to 1.9% of the participants, it is possible to highlight a persistence of symptoms even after recovery from COVID-19 in a percentage of children ranging from 20% to 40%. However, according to most pediatricians (97.3%), the persistence of symptoms is found in less than 20% of children.

Specifically, with regard to the symptoms that persist even after PCR negativization, we recorded the following responses (one or more responses were allowed in the questionnaire): (1) fatigue 75.6%, (2) headache 17.1%, (3) gastrointestinal disorders 16.2%, (4) anosmia 14.5%, (5) cough/dysponea/chest tightness 12.8%, (6) ageusia 9.8%, (7) low-grade fever 8.5%, and (8) vasculitis 4.7%. The absolute numbers of the responses are presented in Figure 1.

Furthermore, it can be seen that only 4.6% of pediatricians performed the antibody titer dosing after clinical recovery; according to 18.3% of the interviewees it depends on the cases, while the majority (77.2%) do not perform the antibody titer assay in any case. Finally, according to 85.9% of pediatricians, in their geographical area, no assistance is dedicated to the management of children who have recovered from COVID-19.

## 4. Discussion

The results of our survey contribute to the description and quantification of long-COVID in the pediatric population, a phenomenon for which there is still little evidence in the literature on the actual occurrence. The first fact that stands out is that unlike what happens in adults in whom the percentage of patients who complain of the persistence of symptoms after recovery from COVID-19 is 80% [3], in our pediatric cases, this persistence does not seem to exceed 20%. Regarding the most reported symptoms, in first place fatigue has been reported followed by headache and gastrointestinal disorders. Taste and smell disturbances and other symptoms seem to have little relevance in this population.

The few reports in the literature on this particular subject are not always in agreement but obviously they are difficult to compare due to the different methods of evaluation and quantification. Buonsenso et al. examined a cohort of 129 children younger than 18 years who had been diagnosed with COVID-19 and were evaluated during the first and second pandemic waves. Most of them had mild COVID-19 symptoms at the time of diagnosis, and 33 were asymptomatic. Of the initial group, 68 were evaluated even after 120 days, and 51% still reported at least one symptom. The most frequent symptoms were fatigue, muscle or joint pain, headache, sleep disturbances, chest pain, dyspnea, and palpitations. On average, five months after diagnosis, only 42% had fully recovered and about one in three children still had at least one symptom [8]. Ludvigsson evaluated five children over time who had COVID-19, demonstrating that four out of five continued to have headaches, concentration disorders, and/or muscle weakness even six months after recovering from COVID-19 [7]. These data seem to have been confirmed by the same Swedish author who described a larger case series involving 35 children [10]. According to a Dutch survey involving 78% of national pediatric departments, 89 children had clinical characteristics attributable to long-covid. Specifically, the main complaints were fatigue, dyspnea and difficulty concentrating [9]. Molteni et al. evaluated symptom prevalence, burden and illness duration in 1734 children with COVID-19. The prevailing symptoms were headache (62.2%) and fatigue (55.0%). Median illness duration was 6 days with a positive correlation with age. Only 77 children (4.4%) had illness duration for more than 28 days. These latter children had a median of six symptoms in the first week, most commonly fatigue (84.4%), headache (77.9%) and anosmia (77.9%). However, after day 28, the symptom burden fell (median 2 symptoms). Authors concluded that some children who tested positive for SARS-CoV-2 experience prolonged illness duration. Reassuringly, symptom burden in these children did not increase with time, and most recovered by day 56 [12].

Different numbers come from another recent study that prospectively evaluated 90 children with persistent symptoms who presented to a designated multidisciplinary clinic for long COVID. In nearly 60%, symptoms were associated with functional impairment at 1–7 months after the onset of infection [13].

Focusing on the data of our survey, it would therefore seem that the phenomenon of long-COVID in pediatric age is overestimated since <20% of Italian pediatricians have reported this phenomenon. This finding confirms that children do not undergo an acute course of the disease that is much more attenuated than adults but are also more protected against long-term complications.

Regarding the symptoms most frequently encountered, fatigue is in first place. Manifestations of long-COVID mimic those of chronic fatigue syndrome (CFS), which includes the presence of severe incapacitating asthenia, pain, neurocognitive impairment, sleep disturbances, autonomic dysfunction, and worsening of overall symptoms following even mild physical or mental exertion [14,15].

CFS is currently a complex and controversial clinical entity with no well-defined causal factors [15,16]. Possible causes include viruses, immune dysfunction, metabolic endocrinological changes, and neuropsychiatric factors [14]. Infectious agents that have been related to CFS include Epstein-Barr virus, cytomegalovirus, enterovirus, herpesvirus [17], and SARS-CoV-1 [18,19] (which is very similar to SARS-CoV-2). It is certainly tempting to speculate that SARS-CoV-2 could be added to the list of viral agents causing CFS. Lidbury et al. proposed a hypothesis to explain the possible mechanisms underlying chronic post-viral fatigue syndrome; inflammatory cytokines abundantly produced during acute disease would impact on the functionality of mechanistic target of rapamycin (mTOR), a serine/threonine kinase that regulates cellular homeostasis by influencing transcription, protein synthesis, autophagy, metabolism, biogenesis, and maintenance of organelles through its signaling pathways [20,21]. Among its many functions, mTOR has an important role in the regulation of mitochondrial activity, oxygen consumption, and oxidative capacity, which are also based on the nutritional status and energy needs of the cell [22]. Therefore, the altered functionality of mTOR caused by inflammatory cytokines would impact mitochondrial function and cellular energy regulation possibly leading to chronic fatigue.

Returning to our survey, the strengths of the survey are represented above all by the high number of pediatricians who completed the questionnaire and that they were above all primary care pediatricians (the first pediatricians with whom families interface for routine problems). Our study also has limitations. The biggest limitation is that this survey is not an objective assessment of symptoms in children with previous COVID-19. Moreover, we did not evaluate sleeping disturbances or concentrations disorders and most of the pediatricians were from the same age group. Furthermore, it is not possible to draw conclusions or make correlations with pre-existing risk factors, duration and severity of the disease, or age of the children. Finally, although the phenomenon of long-COVID in the pediatric population would seem to have little relevance, it is interesting to highlight that almost 86% of pediatricians who answered the questionnaires report that in their area, no reference center dedicated to the assistance of the child recovering from COVID can be found. In this sense, further studies are necessary to determine the burden of long-COVID in the pediatric population and to define the need for centers dedicated to the management of this entity from a clinical care point-of-view.

## 5. Conclusions

This large Italian survey helps to clarify the phenomenon of long-COVID in the pediatric population, which similar to the acute pathology of COVID-19, would seem to be a phenomenon of limited burden in the pediatric population, affecting less than 20% of children. Among all of the symptoms, the one that was most prevalent is fatigue, a pathological entity that is found in many viral diseases.

## Figures and Tables

**Figure 1 children-08-00769-f001:**
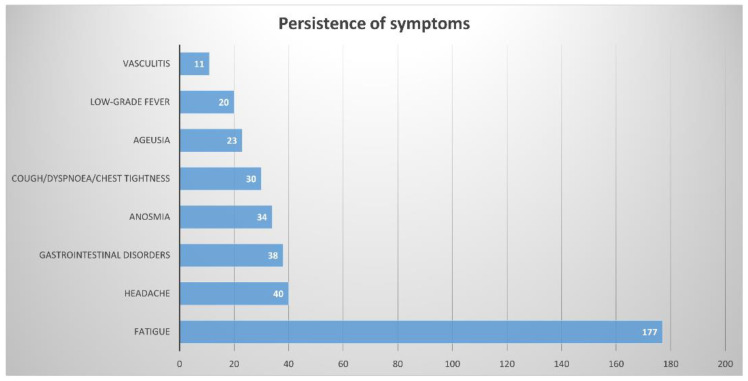
Absolute numbers of answers relating to persistence of symptoms.

**Table 1 children-08-00769-t001:** Questions of the survey covered by this article.

QUESTION	POSSIBLE ANSWERS
When the swab was negative, how many patients were still symptomatic?	a. 0–20% b. 20–40% c. 40–60% d. 60–80% e. >80%
Which of the following symptoms seem to persist even after the swab negativization (one or more responses allowed)?	a. low-grade fever b. cough/dyspnea/chest tightness c. fatigue d. anosmia e. ageusia f. headache g. vasculitis h. gastrointestinal disorders
Is it your habit to evaluate the antibody response with a serological test after clinical recovery?	a. Yes b. No c. It depends on the cases
In your reality, is there a clinic dedicated to the management of children recovered from covid-19?	a. Yes b. No

**Table 2 children-08-00769-t002:** Survey participants’ characteristics.

Total of Participants	267
Sex (male versus female)	92 (34.4%) versus 175 (65.6%)
Age (years old):	
I. 20–30	I. 0 (0%)
II. 31–40	II. 11 (4.2%)
III. 41–50	III. 40 (14.9%)
IV. 51–60	IV. 84 (31.3%)
V. >60	V. 132 (49.6%)
Types of pediatricians:	
**i.** Primary care	I. 202 (75.5%)
**ii.** Pediatric hospital medicine	II. 42 (16.3%)
**iii.** Pediatric emergency medicine	III. 5 (1.9%)
**iv.** Specialized outpatient healthcare	IV. 14 (5.4%)
**v.** Pediatric critical care medicine	V. 2 (0.9%)
**vi.** Pediatric infectious disease	VI. 0%
**vii.** no answer	VII. 2
Territorial subdivisions (South and Islands versus Center of Italy versus Northern Italy)	52 (19.4%) versus 73 (27.4%) versus 142 (53.2%).

## Data Availability

The datasets used and/or analysed during the current study are available from the corresponding author on reasonable request.

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
