# Peer review of "Cross-Sectional Survey on Long Term Sequelae of Pediatric COVID-19 among Italian Pediatricians"

_children, 2021, doi:10.3390/children8090769_

Round 1

Reviewer 1 Report

Main comments: In the present study the authors performed a cross-sectional survey on long term sequelae of pediatric COVID-19 among 267 Italian pediatricians. Most pediatricans reported persistence of symptoms in less than 20% of their patients and fatigue was the most prevalent clinical persistent symptom. This brief report is of clinical interest, however the presentation of the data needs to be improved.

1.) In the abstract the authors state that they showed only some of the data of this survey. I recommend that they provide not only some data but all data of this survey, as only limited data are reported.

2.) The first paragraph of the introduction needs at least two citations and reference 1 is not cited in the text

3.) Did the authors get ethical approval for this study?

4.) The main results of the study (persistence of symptoms) are only reported as percentages in the text. Not only the percentages but also absolute numbers and missing values must be given and I recommend to present the data in at least 1 Table or 1 Figure.

5.) In table 2 (survey of participants characteristics) not only the percentages but also the absolute numbers and missing values should be given.

6.) In the discussion Line 177 they authors claim that all participants were primary care pediatricans, although table 2 shows that only 75.5% were primary care pediatricans. What is the correct message?

7.) Further important limitations not mentioned by the authors were that they did not evaluate persistent symptoms such as sleeping disturbances and concentrations disorders and they did not evaluate the severity of SARS CoV-2 infections at time of diagnosis. I recommend to include those limitations.

Minor comments:

1.) Abstract Line 26: change "has occurs" to "had occured"
Line 36 change "the one that is most prevalent" to "the one that was most prevalent"

2.) Results Line 111: do not use the term "after healing". If the patient is still symptomatic. You could use after PCR negativization.

3.) Table 2: change "ediatric infectious disease" to "pediatric infectious disease"

4.) Line 152: change "are also be more protected against" to " are also more protected against"

5.) Line 194: change "the one that is prevalent is fatigue" to "the one that was most prevalent is fatigue"

Author Response

REVIEWER #1: 

Main comments: In the present study the authors performed a cross-sectional survey on long term sequelae of pediatric COVID-19 among 267 Italian pediatricians. Most pediatricans reported persistence of symptoms in less than 20% of their patients and fatigue was the most prevalent clinical persistent symptom. This brief report is of clinical interest, however the presentation of the data needs to be improved.

Answer: Thank you for your respectable comments. I hope to provide you all the requested clarifications to improve the manuscript.

1.) In the abstract the authors state that they showed only some of the data of this survey. I recommend that they provide not only some data but all data of this survey, as only limited data are reported.

Answer: Thank you. The entire survey is the subject of another article and focuses on the impact of the pandemic among Italian pediatricians. This article deliberately wants to focus only on the aspect of long-covid in pediatric age since it is an aspect that needs to be explored with further studies; for this reason, we report in this circumstance only the answers relating to this topic since the rest would be off topic. To follow your suggestion as a reviewer, we have decided to add some clarification in the methods section.

2.) The first paragraph of the introduction needs at least two citations and reference 1 is not cited in the text

Answer: Thank you. We added the reference # 1 after the first paragraph.  

3.) Did the authors get ethical approval for this study?

Answer: Thank you. The Ethics Committee of the University of Catania (Italy), was contacted and no special permission was deemed to be required because the study design satisfied the criteria of an activity audit. We added this information in the text.

4.) The main results of the study (persistence of symptoms) are only reported as percentages in the text. Not only the percentages but also absolute numbers and missing values must be given and I recommend to present the data in at least 1 Table or 1 Figure.

Answer: Thank you for your excellent suggestion. We added the figure 1 representing the absolute numbers.

5.) In table 2 (survey of participants characteristics) not only the percentages but also the absolute numbers and missing values should be given.

Answer: Thank you for your excellent suggestion. We added the absolute numbers. 

6.) In the discussion Line 177 they authors claim that all participants were primary care pediatricans, although table 2 shows that only 75.5% were primary care pediatricans. What is the correct message?

Answer: Thank you. They were “above all” primary care pediatricians. We corrected the text.  

7.) Further important limitations not mentioned by the authors were that they did not evaluate persistent symptoms such as sleeping disturbances and concentrations disorders and they did not evaluate the severity of SARS CoV-2 infections at time of diagnosis. I recommend to include those limitations.

Answer: Thank you. We added those limitations.  

Minor comments:

1.) Abstract Line 26: change "has occurs" to "had occurred"

Line 36 change "the one that is most prevalent" to "the one that was most prevalent"

Answer: Thank you. We corrected accordingly.   

2.) Results Line 111: do not use the term "after healing". If the patient is still symptomatic. You could use after PCR negativization.

Answer: Thank you. As you suggested we modified the sentence.   

3.) Table 2: change "ediatric infectious disease" to "pediatric infectious disease"

Answer: Thank you. As you suggested, we corrected the typo.

4.) Line 152: change "are also be more protected against" to " are also more protected against"

Answer: Thank you. As you suggested we modified the sentence.

5.) Line 194: change "the one that is prevalent is fatigue" to "the one that was most prevalent is fatigue"

Answer: Thank you. As you suggested we modified the sentence.

Reviewer 2 Report

The authors show us the results of a survey of Italian Pediatricians on their perception of Long COVID in children. More than 600 authors express their opinions. This work is not entirely new, since the authors already carried out a previous survey with a smaller number of participants.
As they already mentioned, when dealing with perceptions, there is no real confirmation of the percentages of Long COVID and symptoms in the pediatric population, this being the main limitation. An actual survey could yield different results.
Other limitations are that almost all pediatricians are from Primary Care and from a specific age group.
However, it is interesting to draw attention to this still a very little known disease in children.

As specific comments, I would like to know how many pediatricians the survey was sent to, and how many did not answer.
Among the limitations, it should be included that, curiously, all pediatricians are of the same age group (perhaps they have more time) and that they are from Primary Care. Hsoptiary pediatricians could have another perception.

Author Response

REVIEWER #2: 

The authors show us the results of a survey of Italian Pediatricians on their perception of Long COVID in children. More than 600 authors express their opinions. This work is not entirely new, since the authors already carried out a previous survey with a smaller number of participants.
As they already mentioned, when dealing with perceptions, there is no real confirmation of the percentages of Long COVID and symptoms in the pediatric population, this being the main limitation. An actual survey could yield different results.
Other limitations are that almost all pediatricians are from Primary Care and from a specific age group.
However, it is interesting to draw attention to this still a very little known disease in children.

 Answer: Thank you for your respectable comments. I hope to provide you all the requested clarifications to improve the manuscript.

As specific comments, I would like to know how many pediatricians the survey was sent to, and how many did not answer.

 Answer: Thank you. The questionnaire was sent to about 500 pediatricians. Thus, we have achieved more than 50% of the response rate. Following your suggestion, we added this information in the text.

Among the limitations, it should be included that, curiously, all pediatricians are of the same age group (perhaps they have more time) and that they are from Primary Care. Hsoptiary pediatricians could have another perception.

Answer: Thank you. In Italy, there are nearly 10,000 primary care pediatricians and about 6,000 hospital pediatricians. The proportion between the two components is more or less respected. Furthermore, primary care pediatricians are probably the ones who know best the long-term issues, as they are the ones that do not require hospitalization. We believe this represents a strength and not a limitation. As for the age factor, this too reflects the local reality: most pediatricians are over 50 years old. As you have suggested, we have added this limitation in the text. Thanks.

Round 2

Reviewer 1 Report

The authors improved their manuscript and considered all reviewer comments in the revised manuscript.

Reviewer 2 Report

The authors have done the suggested modifications.